# MAP: Unleashing Hybrid Mamba-Transformer Vision Backbone's Potential with Masked Autoregressive Pretraining

## Abstract

Mamba has achieved significant advantages in long-context modeling and autoregressive tasks, but its scalability with large parameters remains a major limitation in vision applications. pretraining is a widely used strategy to enhance backbone model performance. Although the success of Masked Autoencoder in Transformer pretraining is well recognized, it does not significantly improve Mamba's visual learning performance. We found that using the correct autoregressive pretraining can significantly boost the performance of the Mamba architecture. Based on this analysis, we propose Masked Autoregressive Pretraining(MAP) to pretrain a hybrid Mamba-Transformer vision backbone network. This strategy combines the strengths of both MAE and Autoregressive pretraining, improving the performance of Mamba and Transformer modules within a unified paradigm. Experimental results show that both the pure Mamba architecture and the hybrid Mamba-Transformer vision backbone network pretrained with MAP significantly outperform other pretraining strategies, achieving state-of-the-art performance. We validate the effectiveness of the method on both 2D and 3D datasets and provide detailed ablation studies to support the design choices for each component.

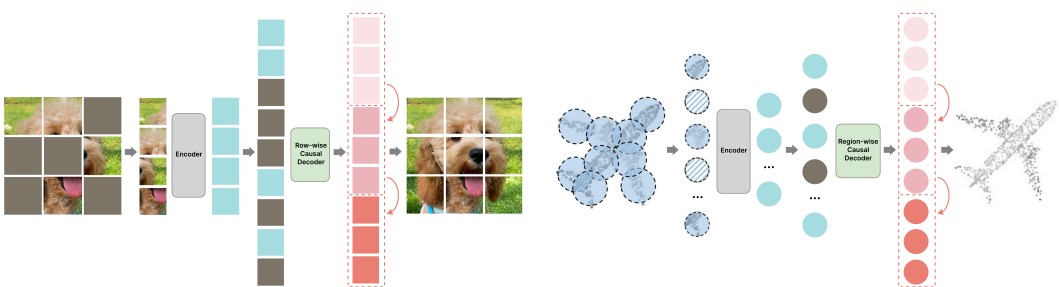

Figure 1: We propose Masked Autoregressive Pretraining(MAP) to pretrain the hybrid Mamba-Transformer vision backbones. This strategy combines the strengths of both MAE and Autoregressive, improving the performance of Transformer and Mamba modules within a unified paradigm.

## 1 Introduction

The State Space Model(Hamilton, 1994) has demonstrated strong capabilities in long-context language modeling. The recent emergence of the variant framework Mamba(Gu & Dao, 2023) has sparked interest in comparing its abilities with those of Transformers. Due to its linear complexity and selective scanning mechanism, Mamba shows significant advantages in computational efficiency when handling long contexts. However, Mamba-based architectures(Zhu et al., 2024b) are difficult to scale concerning the number of parameters, which poses a major limitation for vision applications. To enhance Mamba-based backbones for vision tasks, there's a trend of combining Mamba with Transformers to create hybrid backbones(Lieber et al., 2024; Hatamizadeh & Kautz, 2024), leveraging the strengths of both. However, to truly scale up these hybrid vision backbones,

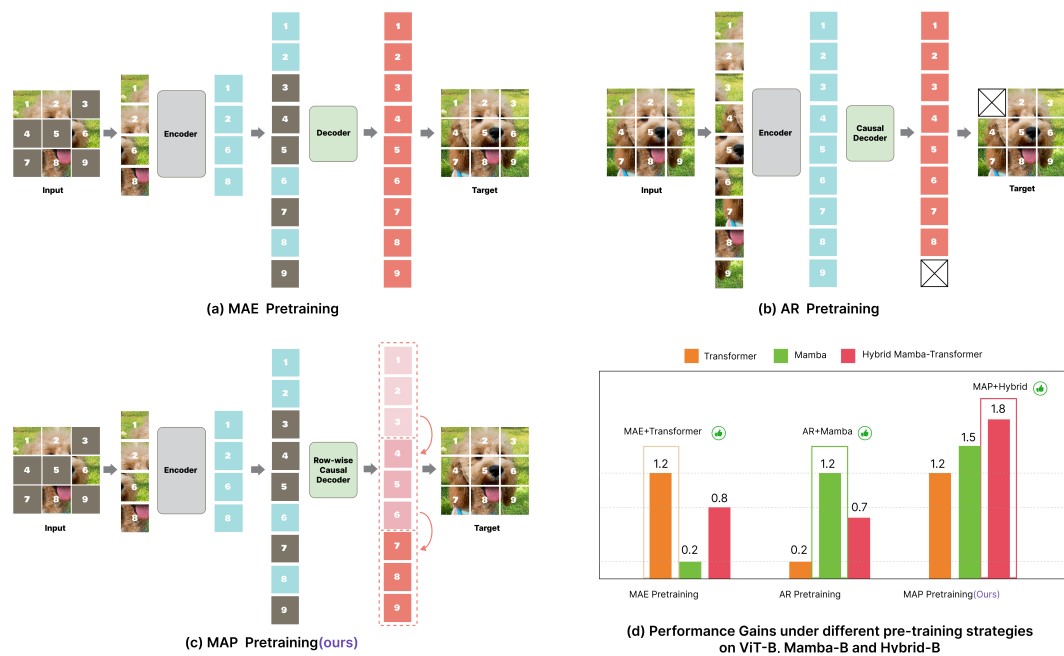

Figure 2: **(a) MAE Pretraining.** Its core lies in reconstructing the masked tokens based on the unmasked tokens to build a global bidirectional contextual understanding. **(b) AR Pretraining.** It focuses on building correlations between contexts, and its scalability has been thoroughly validated in the field of large language models. **(c) MAP Pretraining(ours).** Our method first randomly masks the input image, and then reconstructs the original image in a row-by-row autoregressive manner. This pretraining approach demonstrates significant advantages in modeling contextual features of local characteristics and the correlations between local features, making it highly compatible with the Mamba-Transformer hybrid architecture. **(d) Performance Gains under different pretraining strategies on ImageNet-1K.** We found MAE pretraining is better suited for Transformers, while AR is more compatible with Mamba. MAP, on the other hand, is more suited for the Mamba-Transformer backbone. Additionally, MAP also demonstrates impressive performance when pretraining with pure Mamba or pure Transformer backbones, showcasing the effectiveness and broad applicability of our method.

a good pretraining strategy is essential for maximizing the combined capabilities of Mamba and Transformer. Our work aims to take the first step in this direction.

Developing an effective pretraining strategy for Mamba-Transformer vision backbones is challenging. Even for purely Mamba-based backbones, pretraining methods are still underexplored, and the optimal approach remains unclear. Additionally, the hybrid structure requires a pretraining strategy compatible with both computation blocks. This is particularly challenging because the State Space Model captures visual features very differently from Transformers.

To address these challenges, we conducted extensive pilot studies and identified three key observations. Firstly, existing popular pretraining strategies for Transformers, such as MAE(He et al., 2022) and Contrastive Learning(CL)(He et al., 2020), do not yield satisfactory results for Mamba-based backbones, highlighting the need for a more suitable method. Secondly, Autoregressive Pretraining(AR)(Ren et al., 2024) can be effective for Mamba-based vision backbones, provided that an appropriate scanning pattern and token masking ratio are employed. Thirdly, pretraining strategies suitable for either Mamba or Transformers may not effectively benefit the other, and hybrid backbones require a tailored approach to address the learning needs of different computation blocks.

Based on the above observations, we develop a novel pretraining strategy suitable for the Mamba-Transformer vision backbone named Masked Autoregressive pretraining, or MAP for short. The key is a hierarchical pretraining objective where local MAE is leveraged to learn good local attention for the Transformer blocks while global autoregressive pretraining enables the Mamba blocks to learn meaningful contextual information. Specifically, the pretraining method is supported by two key

designs. First, we leverage local MAE to enable the hybrid framework, particularly the Transformer module, to learn local bidirectional connectivity. This requires the hybrid network to predict all tokens within a local region after perceiving local bidirectional information. Second, we autoregressively generate tokens for each local region to allow the hybrid framework, especially the Mamba module, to learn rich contextual information. This requires the network to autoregressively generate subsequent local regions based on the previously decoded tokens.

Our experiments demonstrate that hybrid Mamba-Transformer models pretrained with MAP outperform other pretraining strategies by a significant margin. MAP with the hybrid Mamba-Transformer and pure Mamba backbone can both achieve impressive results on the ImageNet-1k(Deng et al., 2009a) classification task and other 3D vision tasks(Yi et al., 2016; Wu et al., 2015; Uy et al., 2019b). Furthermore, we tried different hybrid integration strategies for combining Mamba and Transformer layers showing that placing Transformer layers at regular intervals within Mamba layers led to a substantial boost in downstream task performance.

Our contributions are threefold:
**Firstly,** we propose a novel method for pretraining the Hybrid Mamba-Transformer Vision Backbone for the first time, enhancing the performance of hybrid backbones as well as pure Mamba and pure Transformer backbones within a unified paradigm.
**Secondly,** we conduct an in-depth analysis of the key components of Mamba with autoregressive pretraining, revealing that the effectiveness hinges on maintaining consistency between the pretraining order and the Mamba scanning order, along with an appropriate token masking ratio.
**Thirdly,** we demonstrate that our proposed method, MAP, significantly improves the performance of both Mamba-Transformer and pure Mamba backbones across various 2D and 3D datasets.

## 2 RELATED WORK

**Vision Mambas and Vision Transformers.** Vision Mamba(Vim)(Zhu et al., 2024a) is an efficient model for visual representation learning, leveraging bidirectional state space blocks to outperform traditional vision transformers like DeiT in both performance and computational efficiency. The VMamba(Liu et al., 2024) architecture, built using Visual State-Space blocks and 2D Selective Scanning, excels in visual perception tasks by balancing efficiency and accuracy. Autoregressive pretraining(ARM)(Ren et al., 2024) further boosts Vision Mamba's performance, enabling it to achieve superior accuracy and faster training compared to conventional supervised models. Nevertheless, why autoregression is effective for Vision Mamba and what the key factors are remains an unresolved question. In this paper, we explore the critical design elements behind the success of Mamba's autoregressive pretraining for the first time. Vision Transformers(ViT)(Dosovitskiy, 2020) adapt transformer architectures to image classification by treating image patches as sequential tokens. Swin Transformer(Liu et al., 2021) introduces a hierarchical design with shifted windows, effectively capturing both local and global information for image recognition. MAE (He et al., 2022) enhances vision transformers through self-supervised learning, where the model reconstructs masked image patches using an encoder-decoder structure, enabling efficient and powerful pretraining for vision tasks. However, the MAE pretraining strategy is not effective for Mamba, which hinders our ability to pretrain the hybrid Mamba-Transformer backbones.

**Self-Supervised Visual Representation Learning.** Self-Supervised Visual Representation Learning is a machine learning approach that enables the extraction of meaningful visual features from large amounts of unlabeled data. This methodology relies on pretext tasks, which serve as a means to learn representations without the need for explicit labels. GPT-style AR(Han et al., 2021) models predict the next part of an image or sequence given the previous parts, encouraging the model to understand the spatial or temporal dependencies within the data. MAE(He et al., 2022) methods mask out random patches of an input image and train the model to reconstruct these masked regions. This technique encourages the model to learn contextual information and global representations. Contrastive Learning(CL)(He et al., 2020) techniques involve contrasting positive and negative samples to learn discriminative features. It typically involves creating pairs of positive and negative examples and training the model to distinguish between them. However, we found that existing pretraining strategies fail to fully unlock the potential of the hybrid framework, which motivated us to explore a new pretraining paradigm for hybrid Mamba-Transformer backbones.

## 3 PILOT STUDY: HOW TO PRE-TRAIN THE VISUAL MAMBA BACKBONES?

In this Section, we first conduct experiments to investigate the differences in pretraining strategies for ViT and Vim. The success of the MAE strategy on the ViT architecture is well acknowledged, while the Vim pretraining strategy remains in its early stages. We are interested in determining whether the MAE strategy is equally applicable to Vim or if the AR strategy is more suitable. To explore this, we conduct experiments on the classification task using the ImageNet-1K dataset. The results are shown in Table 1.

| Method | ViT | ViT+MAE | ViT+AR | ViT+CL |
|--------|-----|---------|--------|--------|
| Accuracy | 82.3 | **83.6(+1.4)** | 82.5(+0.2) | 82.5(+0.2) |
| Method | Vim | Vim+MAE | Vim+AR | Vim+CL |
| Accuracy | 81.2 | 81.4(+0.2) | **82.6(+1.4)** | 81.1(-0.1) |

Table 1: Pilot Study. We use ViT-B and Vim-B as the default configurations. The AR strategy processes the image tokens in a row-first order, while the MAE operates according to the default settings. For contrastive learning, we only used crop and scale data augmentation and used the MoCov2 for pretraining. All experiments are conducted at a resolution of 224x224. The number of mask tokens for AR is set to 40 tokens (20%). Experiments show that MAE is more suitable for Transformer pretraining, while AR is better suited for Mamba pretraining.

We observe that the MAE strategy significantly enhances the performance of ViT. However, for Vim, the MAE strategy does not yield the expected improvements, while the AR strategy substantially boosts its performance. This indicates that for the ViT architecture, applying the MAE strategy is essential to establish bidirectional associations between tokens, thereby improving performance. In contrast, for Vim, it is more important to model the continuity between preceding and succeeding tokens. Based on this observation, we conducted an in-depth analysis of the various components involved in AR pretraining for Mamba and discovered that consistent autoregression pretraining with scanning order and proper masking ratio is the key to pretraining Mamba.

**Relationship between AR and Scanning Order.** Since the goal of AR pretraining is to learn a high-quality conditional probability distribution, enabling the model to generate new sequences based on previously generated content, we first explore how the prediction order in auto-regressive models affects the pretraining of Vim. Different prediction orders can significantly impact how the model captures image features and the effectiveness of sequence generation. By adjusting the prediction order, we can gain deeper insights into Vim's behavior in sequence generation tasks and how to effectively model dependencies between elements in an image. Further analysis of the role of prediction order will help optimize AR pretraining for Vim, exploring how the model can better capture the continuity and relationships of image information under different contextual conditions. We conduct ablation studies on Vim by allowing it to perform both row-first and column-first scanning. We then pretrain it with row-first and column-first AR orders, respectively, to compare their performance. Figure 3 shows different orders for AR pretraining and Mamba scanning.

| Method | Vim(R) | Vim(R) + AR(C) | Vim(R) + AR(R) |
|--------|--------|----------------|----------------|
| Accuracy | 79.7 | 79.9(+0.2) | **82.6(+2.9)** |
| Method | Vim(C) | Vim(C) + AR(C) | Vim(C) + AR(R) |
| Accuracy | 79.5 | **82.5(+3.0)** | 79.9(+0.4) |

Table 2: The impact of AR pretraining order on downstream tasks. Vim(R) refers to Vim with row-first scanning. Vim(C) refers to Vim with column-first scanning. AR(R) refers to row-first autoregressive pretraining. AR(C) refers to column-first autoregressive pretraining. The results indicate that the best performance is achieved when the auto-regressive pretraining design aligns with Mamba's scanning order.

The results are shown in Table 2. We observe that employing a pretraining strategy consistent with the scanning order significantly enhances Vim's performance. This suggests that when designing pretraining strategies, they should be aligned with the downstream scanning order.

**Masking Ratio of Autoregression Pretraining.** Since the success of MAE is primarily attributed to the use of an appropriate masking ratio, we are inspired to conduct experiments to verify whether different auto-regressive masking ratios will affect the quality of pretraining. We found that during AR pretraining, masking a certain number of tokens at the end of the sequence is crucial. Masking a single token follows the traditional AR paradigm, while masking $n$ tokens transforms the task

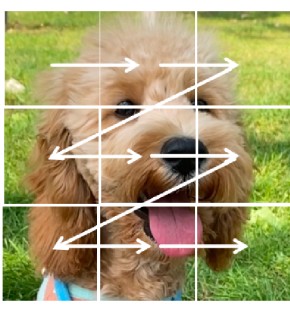 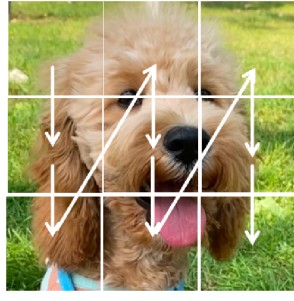

Row First Order      Column First Order

Figure 3: Different orders for AR pretraining and Mamba scanning. The row-first and column-first orders allow the network to perceive local information in different ways and sequences.

into an inpainting problem, as the input and output sequence lengths remain equal. In this context, varying the auto-regressive masking ratios effectively adjusts the inpainting ratio, influencing the model's predictions beyond just the sequence length. Our pretraining sequence length was set to 196 tokens, and we masked 1 token (0.5%), 20 tokens (10%), 40 tokens (20%), 60 tokens (30%), 100 tokens (50%), and 140 tokens (70%), respectively, while also recording the results of fine-tuning on downstream tasks. Figure 4 shows the pipeline of AR Pretraining under different mask ratios.

| Masked tokens | 1 (0.5%) | 20 (10%) | 40 (20%) |
|---|---|---|---|
| Accuracy | 81.7 | 82.0 | **82.6** |
| Masked tokens | 60 (30%) | 100 (50%) | 140 (70%) |
| Accuracy | 82.5 | 82.2 | 81.9 |

Table 3: The impact of Masking Ratio on AR pretraining. We masked 1 token (0.5%), 20 tokens (10%), 40 tokens (20%), 60 tokens (30%), 100 tokens (50%), and 140 tokens (70%), respectively, while also recording the results of fine-tuning on downstream tasks. The experiment shows that an appropriate masking ratio is important for autoregressive pretraining.

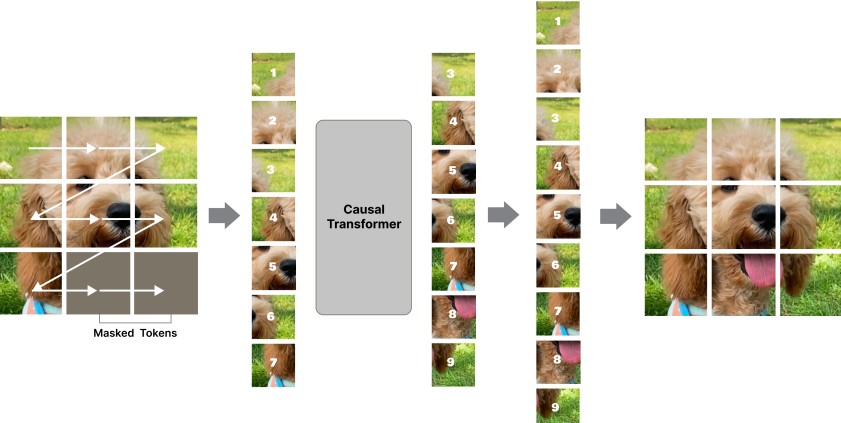

Figure 4: Masking Ratio of Autoregression Pretraining. We showcased the autoregressive training process at various masking ratios. Notably, in autoregressive pretraining, different masking ratios effectively control not only the prediction step size but also the length of the input sequence.

The results shown in Table 3 indicate that a proper masking ratio contributes to training stability, helping to avoid excessive noise interference. In auto-regressive pretraining, as the Masking Ratio increases, the performance of the Mamba improves. This is because a higher Masking Ratio encourages the model to learn more complex and rich feature representations, thereby enhancing its generative ability and adaptability. However, an excessively high Masking Ratio may lead to instability during the training process and result in incomplete information perception. We found there exists a sweet spot around 20% on the ImageNet-1K classification task. In such cases, the model may struggle to make accurate predictions due to a lack of sufficient contextual information, negatively impacting its pretraining effectiveness. Therefore, when designing auto-regressive pretraining tasks, finding an appropriate masking ratio is crucial to strike a balance between performance improvement and training stability.

Given that MAE is more suitable for Transformers while AR is better suited for Mamba, how should we approach the pretraining of a hybrid Mamba-Transformer model? We need a new pretraining strategy that is effective for both Transformers and Mamba to support the pretraining of hybrid models. In the next Section, we will provide a detailed explanation of how to pretrain the hybrid Mamba-Transformer backbones.

## 4 Masked Autoregressive Pretraining for Hybrid Backbones

Our approach represents a general paradigm applicable to data across various domains, with 2D image data as an example. Our method can be easily extended to large language models (LLMs) and the fields of image video and point cloud video. Our method optimizes the synergy between Mamba and Transformer within a unified framework, allowing both models to fully leverage their strengths. In the Mamba-Transformer hybrid architecture, this approach effectively enhances the cooperation between the models, resulting in significant performance improvements. Specifically, our approach includes a masking strategy, a hybrid Mamba-Transformer encoder, and a Transformer decoder. The hybrid Mamba-Transformer encoder is responsible for mapping the signals into latent space, while the Transformer decoder autoregressively reconstructs the features back into the original image. The following section will introduce the specific design components of the framework. The subsequent experiments in this section are conducted using the base-sized model on the ImageNet-1K dataset.

**Masking.** Consistent with MAE, we first tokenize the image and then apply random masking to a portion of the tokens. We experimented with different masking strategies, including random, sequential, and diagonal masking. Our experiments show that random masking delivers the best results. We attribute this to the fact that sequential and diagonal masking can hinder the Transformer's ability to establish contextual relationships. Random masking not only promotes bidirectional modeling for Transformers but also enhances Mamba's generalization and representation capabilities in sequence modeling. Additionally, we explored the effects of different masking ratios and found that a 50% masking ratio yielded the best results. This conclusion aligns with intuition: while MAE performs optimally on Transformers with a 75% masking ratio, previous experiments showed that AR achieves the best results on Mamba with a 20% ratio. Therefore, a 50% ratio serves as a balanced number, leveraging the strengths of both paradigms.

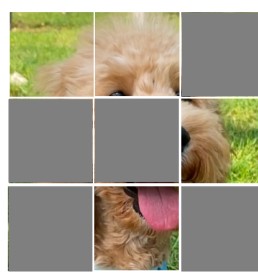 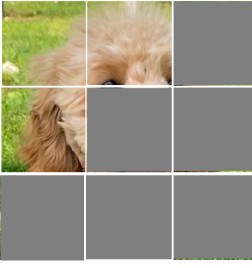

| (a) Random Masking | (b) Sequential Masking | (c) Diagonal Masking |

Figure 5: Different Masking Strategies. The random masking strategy produces the best results.

| Masking Design | From Scratch | Random Masking | Sequential Masking | Diagonal Masking |
|---|---|---|---|---|
| Accuracy | 83.1 | **84.9** | 84.0 | 83.8 |
| Masking Ratio | **0%** | **25%** | **50%** | **75%** |
| Accuracy | 83.3 | 84.5 | **84.9** | 84.2 |

Table 4: Random masking with a 50% masking ratio performs the best.

**MAP Hybrid Mamba-Transformer Encoder.** We designed a series of hybrid Mamba-Transformer vision backbones and compared their performance when trained from scratch. The results indicate that the hybrid approach using MMMTMMMT performs the best. When comparing Mamba-R* with MMMMMMTT, we found that adding a Transformer after Mamba enhances its long-context modeling capabilities, leading to improved performance. However, when comparing MMMMMMTT with TTMMMMMM, we observed that simply appending Transformers after Mamba does not fully leverage the architecture's potential. This suggests that incorporating Transformers at the beginning is crucial for extracting sufficient local features. We believe that the MMMTMMMT approach effectively balances local feature extraction and contextual modeling enhancement, making it our default configuration.

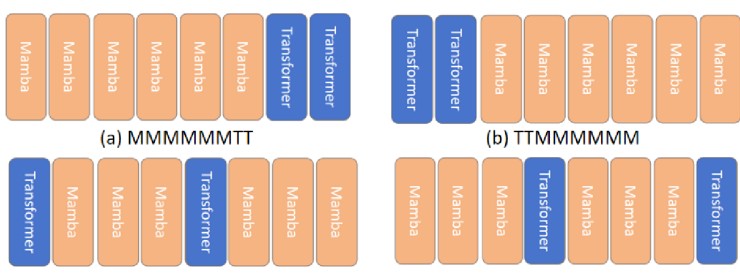

Figure 6: Different Hybrid Model Design. (d) achieves the best results and is set as default.

| Design | DeiT* | Mamba-R* | MMMMMMTT |
|---|---|---|---|
| Accuracy | 82.80 | 82.70 | 82.88 |

| Design | TTMMMMMM | TMMMTMMM | MMMTMMMT |
|---|---|---|---|
| Accuracy | 82.93 | 83.01 | **83.12** |

Table 5: Hybrid Design of Mamba-Transformer backbone. All experiments are trained from scratch. Mamba-R* means 24 Mamba-R(Wang et al., 2024) Mamba layers plus 8 additional Mamba layers. DeiT* means 24 DeiT(Touvron et al., 2021) Transformer layers plus 8 additional Transformer layers. MMMMMMTT represents 24 Mamba layers followed by 8 Transformer layers. TTMMMMMM represents 8 Transformer layers followed by 24 Mamba layers. TMMMTMMM represents a unit consisting of 1 Transformer layer and 3 Mamba layers, repeated 8 times. MMMTMMMT represents a unit of 3 Mamba layers followed by 1 Transformer layer, repeated 8 times.

**MAP Transformer Decoder.** To reconstruct the original image, we utilize a masked Transformer for signal recovery. Our decoder, while consistent with MAE, employs a distinct row-wise decoding strategy that allows autoregressive decoding of one row of tokens at a time, enhancing the network's ability to capture local features and contextual relationships among regions. Experiments show that this method significantly outperforms the original AR, MAE, and local MAE decoding strategies. Notably, in the hybrid framework, local MAE performs comparably to standard MAE, emphasizing the significance of local feature learning. Our MAP method improves local feature modeling while leveraging autoregressive techniques to capture contextual relationships across regions, resulting in superior performance.

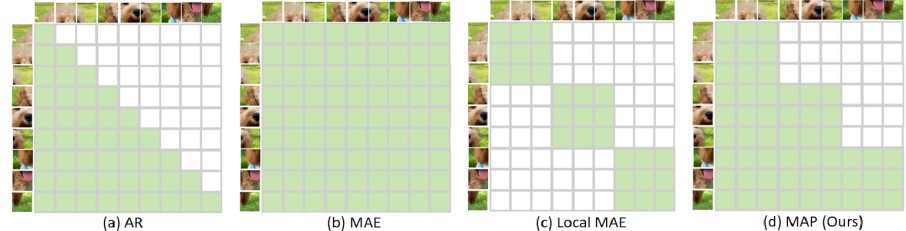

Figure 7: Different Decoder Mask. Green represents activation. White represents non-activation.

| Decoder Mask | Autoregressive(AR) | MAE | local MAE | MAP (ours) |
|---|---|---|---|---|
| Accuracy | 83.7 | 84.1 | 84.2 | **84.9** |

Table 6: Decoder Mask Design. Our MAP decoder strategy achieves the best results.

**Reconstruction Target.** Consistent with MAE, we reconstructed normalized original pixels as the target and employed MSE loss. Inspired by MAR(Li et al., 2024) to use reconstruction output as a conditional signal for diffusion models to improve generation quality, we explored whether pretraining with diffusion loss could enhance performance. However, this approach did not yield significant improvements. This may be due to the decoder's increased capacity negatively impacting the encoder's pretraining effectiveness, suggesting that the quality of reconstructed images is not directly linked to encoder pretraining success.

| Reconstruction Target | From Scratch | Diffusion Loss | MSE Loss (ours) |
|---|---|---|---|
| Accuracy | 83.1 | 83.3 | **84.9** |

Table 7: Reconstruction Target. Results indicate that the quality of the reconstructed image is not directly related to the pretraining effectiveness.

| Model | Img. size | #Params | Throughput | Mem | Acc. (%) |
|---|---|---|---|---|---|
| *Pure Convolutional networks:* | | | | | |
| ResNet-50 (He et al., 2016) | $224^2$ | 25M | 2388 | 6.6G | 76.2 |
| ResNet-152 (He et al., 2016) | $224^2$ | 60M | 1169 | 12.5G | 78.3 |
| EfficientNet-B3 (Tan & Le, 2019) | $300^2$ | 12M | 496 | 19.7G | 81.6 |
| ConvNeXt-T (Liu et al., 2022b) | $224^2$ | 29M | 701 | 8.3G | 82.1 |
| ConvNeXt-S (Liu et al., 2022b) | $224^2$ | 50M | 444 | 13.1G | 83.1 |
| ConvNeXt-B (Liu et al., 2022b) | $224^2$ | 89M | 334 | 17.9G | 83.8 |
| *Pure Vision Transformers:* | | | | | |
| ViT-B/16 (Dosovitskiy et al., 2021) | $224^2$ | 86M | 284 | 63.8G | 77.9 |
| ViT-L/16 (Dosovitskiy et al., 2021) | $224^2$ | 307M | 149 | - | 76.5 |
| *Pretrained Vision Transformers:* | | | | | |
| ViT-B/16 + MAE (Dosovitskiy et al., 2021) | $224^2$ | 86M | 284 | 63.8G | 83.6 |
| ViT-L/16 + MAE (Dosovitskiy et al., 2021) | $224^2$ | 307M | 149 | - | 85.9 |
| ViT-B/16 + MAP | $224^2$ | 86M | 284 | 63.8G | **83.6** |
| ViT-L/16 + MAP | $224^2$ | 307M | 149 | - | **86.1** |
| *Pure Mamba architecture:* | | | | | |
| Vim-T (Zhu et al., 2024a) | $224^2$ | 7M | 1165 | 4.8G | 76.1 |
| Vim-S (Zhu et al., 2024a) | $224^2$ | 26M | 612 | 9.4G | 80.5 |
| MambaR-T (Wang et al., 2024) | $224^2$ | 9M | 1160 | 5.1G | 77.4 |
| MambaR-S (Wang et al., 2024) | $224^2$ | 28M | 608 | 9.9G | 81.1 |
| MambaR-B (Wang et al., 2024) | $224^2$ | 99M | 315 | 20.3G | 82.9 |
| MambaR-L (Wang et al., 2024) | $224^2$ | 341M | 92 | 55.5G | 83.2 |
| *Pretrained Mamba architecture:* | | | | | |
| ARM-B (Mamba+AR) (Ren et al., 2024) | $224^2$ | 85M | 325 | 19.7G | 83.2 |
| ARM-L (Mamba+AR) (Ren et al., 2024) | $224^2$ | 297M | 111 | 53.1G | 84.5 |
| MambaR-B+MAP | $224^2$ | 99M | 315 | 20.3G | **84.0** |
| MambaR-L+MAP | $224^2$ | 341M | 92 | 55.5G | **84.8** |
| *Hybrid 2D convolution + Mamba:* | | | | | |
| VMamba-T (Liu et al., 2024) | $224^2$ | 31M | 464 | 7.6G | 82.5 |
| VMamba-S (Liu et al., 2024) | $224^2$ | 50M | 313 | 27.6G | 83.6 |
| VMamba-B (Liu et al., 2024) | $224^2$ | 89M | 246 | 37.1G | 83.9 |
| *Hybrid 2Dconvolution + Mamba + Transformer architecture: (with down-sampling)* | | | | | |
| MambaVision-T (Hatamizadeh & Kautz, 2024) | $224^2$ | 35M | 1349 | 10.7G | 82.7 |
| MambaVision-S (Hatamizadeh & Kautz, 2024) | $224^2$ | 51M | 1058 | 36.6G | 83.3 |
| MambaVision-B (Hatamizadeh & Kautz, 2024) | $224^2$ | 97M | 826 | 50.8G | 84.2 |
| MambaVision-L (Hatamizadeh & Kautz, 2024) | $224^2$ | 241M | 229 | 78.6G | 85.3 |
| *Hybrid Mamba + Transformer architecture: (without down-sampling)* | | | | | |
| HybridMH-T | $224^2$ | 12M | 910 | 7.6G | 77.7 |
| HybridMH-S | $224^2$ | 37M | 512 | 14.6G | 81.3 |
| HybridMH-B | $224^2$ | 128M | 244 | 30.0G | 83.1 |
| | $384^2$ | 128M | 244 | 76.1G | 84.5 |
| HybridMH-L | $224^2$ | 443M | 63 | 78.3G | 83.2 |
| | $384^2$ | 443M | 63 | - | 84.6 |
| *Pretrained Hybrid architecture:* | | | | | |
| HybridMH-T + MAP | $224^2$ | 12M | 910 | 7.6G | **78.6** |
| HybridMH-S + MAP | $224^2$ | 37M | 512 | 14.6G | **82.5** |
| HybridMH-B + MAE | $224^2$ | 128M | 244 | 30.0G | 83.9 |
| HybridMH-B + AR | $224^2$ | 128M | 244 | 30.0G | 83.8 |
| HybridMH-B + CL | $224^2$ | 128M | 244 | 30.0G | 83.1 |
| HybridMH-B + MAP | $224^2$ | 128M | 244 | 30.0G | **84.9** |
| | $384^2$ | 128M | 244 | 76.1G | **85.5** |
| HybridMH-L + MAP | $224^2$ | 443M | 63 | 78.3G | **85.0** |
| | $384^2$ | 443M | 63 | - | **86.2** |

Table 8: ImageNet-1k classification results. The throughput is computed on an A100 GPU. The memory overhead is measured with a batch size of 128 on single GPU. Our results are highlighted in  blue . Our proposed MAP method significantly improves the performance of the hybrid Mamba-Transformer backbones. Additionally, we verified that our MAP method also significantly improves the performance of both the pure Mamba framework and the pure Transformer backbone. Our MAP method also significantly outperforms MAE, AR, and CL pretraining on hybrid networks.

| Hybrid Ratio | 3M1T | 3M1T+MAP | 1M3T | 1M3T+MAP | 2M2T | 2M2T+MAP |
|---|---|---|---|---|---|---|
| Accuracy | 83.1 | 84.9 | 83.3 | 85.1 | 83.5 | 84.9 |

Table 9: Results on Different Hybrid Ratio. 3M1T denotes a ratio of 3:1 for Mamba and Transformer, while 3M1T+MAP indicates that it undergoes MAP pretraining first. The results reveal minimal performance differences among the various hybrid ratios after pertaining. Considering computational efficiency and memory savings, we use the 3:1 hybrid ratio as our default configuration.

## 5 EXPERIMENTS

### 5.1 2D EXPERIMENTS ON IMAGENET-1K CLASSIFICATION TASK

**Settings.** We pretrained on the training set of the ImageNet-1K(Deng et al., 2009b) dataset and then fine-tuned on its classification task. We report the top-1 validation accuracy of a single 224x224 crop, and in some settings, we also report the results for a 384x384 crop. During the pretraining phase, we applied a random masking strategy with a 50% masking ratio, using only random cropping as the data augmentation strategy. We utilized AdamW as the optimizer and trained for 1600 epochs across all settings. Additionally, we pretrained using the MAP paradigm on pure Mamba and pure Transformer networks, demonstrating that this paradigm is effective for both frameworks. In the fine-tuning phase, we directly fine-tune for 400 epochs and report the results.

**Results.** Results are shown in Table 8. The results indicate that the hybrid framework achieves a balance between performance and computational overhead. However, simply training the hybrid architecture from scratch does not lead to significant performance improvements compared to pure Mamba and Transformer backbone. Our proposed pretraining method significantly enhances the performance of the hybrid Mamba-Transformer framework. Additionally, we verified that our MAP method also significantly improves the performance of both the pure Mamba framework and the pure Transformer backbone. Furthermore, when comparing models of the base size with other pretraining methods, we observed that contrastive learning pretraining does not yield performance improvements. The original MAE and AR methods also fail to fully exploit the capabilities of the hybrid Mamba-Transformer backbone, with their results significantly lower than our MAP pretraining method. This further demonstrates the effectiveness of our method for the hybrid framework.

**Results with Different Hybrid Ratio for Mamba and Transformer.** In our experiments, we used a 3:1 hybrid ratio of Mamba to Transformer. We also explored other hybrid ratios, and the results, as shown in Table 9, indicate that there are no significant performance differences among the hybrid models with varying ratios after MAP pretraining. Considering computational efficiency and memory savings, we opted to adopt the 3:1 hybrid ratio as our default configuration.

### 5.2 3D EXPERIMENTS ON MODELNET40, SCANOBJECTNN AND SHAPENETPART

**Settings.** We pretrained using the ShapeNet(Chang et al., 2015) dataset, employing random rotation and translation scaling as data augmentation techniques. Each point cloud consists of 1024 points and is divided into 64 patches, with each patch containing 32 points. We also used a hybrid ratio of Mamba to Transformer at 3:1, randomly masking 50% of the patches. Since point clouds are unordered, the concept of rows does not apply here; instead, we randomly generate 32 patches each time and complete the reconstruction process in an autoregressive manner. Similar to Mamba3DHan et al. (2024), we did not adopt any special sorting strategies but ensured that the order of pretraining matches that of the actual Mamba scans. We conducted pretraining on both the hybrid framework and the original Mamba3D to validate their performance advantages in both the pure Mamba framework and the hybrid framework. During pretraining and downstream fine-tuning, we employed the AdamW optimizer with a cosine decay strategy for 300 epochs. For the ModelNet40Wu et al. (2015) fine-tuning experiments, we used translation and scaling as data augmentation, while on ScanObjectNNUy et al. (2019a), we applied random rotation as data augmentation. Additionally, I also performed experiments in few-shot settings and on ShapeNet part(Yi et al., 2016) segmentation.

**Results.** The experiments demonstrate that our method significantly enhances the performance of both the hybrid framework and the pure Mamba framework on 3D tasks. This suggests that our approach can be easily adapted to other domains and data types, such as LLMs and video data. Notably, in the part segmentation task, the performance of the hybrid framework trained from scratch is inferior to that of the pure Mamba framework. However, after pretraining, the advantages of the hybrid framework are fully realized, significantly surpassing the performance of the pure Mamba framework. This further proves that our method can simultaneously harness the potential of both Mamba and Transformer to achieve better performance.

| Method | PT | #P↓ | #F↓ | ScanObjectNN | | | ModelNet40 |
|---|---|---|---|---|---|---|---|
| | | | | OBJ_BG↑ | OBJ_ONLY↑ | PB_T50_RS↑ | 1k P↑ |
| *Supervised Learning Only: Dedicated Architectures* | | | | | | | |
| PointNet(Qi et al., 2017a) | × | 3.5 | 0.5 | 73.3 | 79.2 | 68.0 | 89.2 |
| PointNet++(Qi et al., 2017b) | × | 1.5 | 1.7 | 82.3 | 84.3 | 77.9 | 90.7 |
| DGCNN(Wang et al., 2019) | × | 1.8 | 2.4 | 82.8 | 86.2 | 78.1 | 92.9 |
| PointCNN(Li et al., 2018) | × | 0.6 | - | 86.1 | 85.5 | 78.5 | 92.2 |
| DRNet(Qiu et al., 2021) | × | - | - | - | - | 80.3 | 93.1 |
| SimpleView(Goyal et al., 2021) | × | - | - | - | - | 80.5±0.3 | 93.9 |
| GBNet(Qiu et al., 2022) | × | 8.8 | - | - | - | 81.0 | 93.8 |
| PRA-Ne(Cheng et al., 2021) | × | - | 2.3 | - | - | 81.0 | 93.7 |
| MVTN(Hamdi et al., 2021) | × | 11.2 | 43.7 | 92.6 | 92.3 | 82.8 | 93.8 |
| PointMLP(Ma et al., 2022) | × | 12.6 | 31.4 | - | - | 85.4±0.3 | 94.5 |
| PointNeXt(Qian et al., 2022) | × | 1.4 | 3.6 | - | - | 87.7±0.4 | 94.0 |
| P2P-HorNet(Wang et al., 2022) | ✓ | - | 34.6 | - | - | 89.3 | 94.0 |
| DeLA(Chen et al., 2023) | × | 5.3 | 1.5 | - | - | 90.4 | 94.0 |
| *Supervised Learning Only: Transformer or Mamba-based Models* | | | | | | | |
| Transformer | × | 22.1 | 4.8 | 79.86 | 80.55 | 77.24 | 91.4 |
| PCT(Guo et al., 2021) | × | 2.9 | 2.3 | - | - | - | 93.2 |
| PointMamba | × | 12.3 | 3.6 | 88.30 | 87.78 | 82.48 | - |
| PCM(Zhang et al., 2024) | × | 34.2 | 45.0 | - | - | 88.10±0.3 | 93.4±0.2 |
| SPoTr(Park et al., 2023) | × | 1.7 | 10.8 | - | - | 88.60 | - |
| PointConT(Liu et al., 2023) | × | - | - | - | - | 90.30 | 93.5 |
| Mamba3d w/o vot. | × | 16.9 | 3.9 | 92.94 | 92.08 | 91.81 | 93.4 |
| Mamba3d w/ vot. | × | 16.9 | 3.9 | 94.49 | 92.43 | 92.64 | 94.1 |
| HybridMT3D w/o vot. | × | 19.3 | 4.4 | 92.81 | 92.28 | 91.97 | 93.5 |
| HybridMT3D w/ vot. | × | 19.3 | 4.4 | 94.50 | 92.58 | 92.66 | 94.3 |
| *With Self-supervised pretraining* | | | | | | | |
| Transformer | *OcCo* | 22.1 | 4.8 | 84.85 | 85.54 | 78.79 | 92.1 |
| Point-BERT | *IDPT* | 22.1+1.7[†] | 4.8 | 88.12 | 88.30 | 83.69 | 93.4 |
| MaskPoint | *MaskPoint* | 22.1 | 4.8 | 89.30 | 88.10 | 84.30 | 93.8 |
| PointMamba | *Point-MAE* | 12.3 | 3.6 | 90.71 | 88.47 | 84.87 | - |
| Point-MAE | *IDPT* | 22.1+1.7[†] | 4.8 | 91.22 | 90.02 | 84.94 | 94.4 |
| Point-M2AE | *Point-M2AE* | 15.3 | 3.6 | 91.22 | 88.81 | 86.43 | 94.0 |
| Mamba3d w/o vot. | *Point-BERT* | 16.9 | 3.9 | 92.25 | 91.05 | 90.11 | 94.4 |
| Point-MAE | *Point-MAE* | 22.1 | 4.8 | 90.02 | 88.29 | 85.18 | 93.8 |
| Mamba3d w/o vot. | *Point-MAE* | 16.9 | 3.9 | 93.12 | 92.08 | 92.05 | 94.7 |
| Mamba3d w/ vot. | *Point-MAE* | 16.9 | 3.9 | 95.18 | 94.15 | 93.05 | 95.4 |
| Mamba3d w/o vot. | *MAP* | 16.9 | 3.9 | 93.62 | 92.75 | 92.65 | 95.1 |
| Mamba3d w/ vot. | *MAP* | 16.9 | 3.9 | 95.64 | 94.87 | 93.76 | 95.6 |
| HybridMT3D w/o vot. | *MAP* | 19.3 | 4.4 | 93.88 | 93.03 | 92.95 | 95.4 |
| HybridMT3D w/ vot. | *MAP* | 19.3 | 4.4 | 95.84 | 94.97 | 93.87 | 95.9 |

Table 10: Results on 3D classification tasks. Our results are highlighted in  blue .

| Method | 5-way | | 10-way | |
|---|---|---|---|---|
| | 10-shot↑ | 20-shot↑ | 10-shot↑ | 20-shot↑ |
| *Supervised Learning Only* | | | | |
| DGCNN (Wang et al., 2019) | 31.6 ±2.8 | 40.8 ±4.6 | 19.9 ±2.1 | 16.9 ±1.5 |
| Transformer (Vaswani et al., 2017) | 87.8 ±5.2 | 93.3 ±4.3 | 84.6 ±5.5 | 89.4 ±6.3 |
| Mamba3D (Han et al., 2024) | 92.6 ±3.7 | 96.9 ±2.4 | 88.1 ±5.3 | 93.1 ±3.6 |
| HybridMT3D | 92.8 ±3.2 | 97.0 ±1.9 | 88.4 ±4.3 | 93.1 ±3.8 |
| *with Self-supervised pretraining* | | | | |
| DGCNN+*OcCo*(Wang et al., 2021) | 90.6 ±2.8 | 92.5 ±1.9 | 82.9 ±1.3 | 86.5 ±2.2 |
| OcCo (Wang et al., 2021) | 94.0 ±3.6 | 95.9 ±2.7 | 89.4 ±5.1 | 92.4 ±4.6 |
| PointMamba (Liang et al., 2024) | 95.0 ±2.3 | 97.3 ±1.8 | 91.4 ±4.4 | 92.8 ±4.0 |
| MaskPoint (Liu et al., 2022a) | 95.0 ±3.7 | 97.2 ±1.7 | 91.4 ±4.0 | 93.4 ±3.5 |
| Point-BERT (Yu et al., 2022) | 94.6 ±3.1 | 96.3 ±2.7 | 91.0 ±5.4 | 92.7 ±5.1 |
| Point-MAE (Pang et al., 2022) | 96.3 ±2.5 | 97.8 ±1.8 | 92.6 ±4.1 | 95.0 ±3.0 |
| Mamba3d+*P-B* (Yu et al., 2022) | 95.8 ±2.7 | 97.9 ±1.4 | 91.3 ±4.7 | 94.5 ±3.3 |
| Mamba3d+*P-M* (Pang et al., 2022) | 96.4 ±1.2 | 98.2 ±1.2 | 92.4 ±4.1 | 95.2 ±2.9 |
| Mamba3d+*MAP* | 97.1 ±3.1 | 98.7 ±1.3 | 92.8 ±3.1 | 95.8 ±3.1 |
| HybridMT3D+*MAP* | 97.3 ±2.8 | 98.7 ±0.8 | 93.0 ±3.6 | 96.0 ±2.7 |

| Method | mIoU$_C$ (%)↑ | mIoU$_I$ (%)↑ | #P↓ | #F↓ |
|---|---|---|---|---|
| *Supervised Learning Only* | | | | |
| PointNet (Qi et al., 2017a) | 80.4 | 83.7 | 3.6 | 4.9 |
| PointNet++ (Qi et al., 2017b) | 81.9 | 85.1 | 1.0 | 4.9 |
| DGCNN (Wang et al., 2019) | 82.3 | 85.2 | 1.3 | 12.4 |
| Transformer (Vaswani et al., 2017) | 83.4 | 85.1 | 27.1 | 15.5 |
| Mamba3D(Han et al., 2024) | 83.7 | 85.7 | 23.0 | 11.8 |
| HybridMT3D | 83.5 | 85.6 | 25.1 | 12.9 |
| *with Self-supervised pretraining* | | | | |
| OcCo (Wang et al., 2021) | 83.4 | 84.7 | 27.1 | - |
| PointContrast (Xie et al., 2020) | - | 85.1 | 37.9 | - |
| CrossPoint (Afham et al., 2022) | - | 85.5 | - | - |
| Point-MAE (Pang et al., 2022) | 84.2 | 86.1 | 27.1 | 15.5 |
| PointMamba (Liang et al., 2024) | 84.4 | 86.0 | 17.4 | 14.3 |
| Point-BERT (Yu et al., 2022) | 84.1 | 85.6 | 27.1 | 10.6 |
| Mamba3d+*P-B* (Yu et al., 2022) | 84.1 | 85.7 | 21.9 | 9.5 |
| Mamba3d+*P-M* (Pang et al., 2022) | 84.3 | 85.8 | 23.0 | 11.8 |
| Mamba3d+*MAP* | 84.5 | 86.0 | 23.0 | 11.8 |
| HybridMT3D+*MAP* | 84.7 | 86.3 | 25.1 | 12.9 |

Table 11: (Left) Few-shot classification on ModelNet40 dataset. (Right) Part segmentation on ShapeNetPart dataset. Our results are highlighted in  blue .

## 6 CONCLUSION

In this paper, we begin with an in-depth analysis of the key factors that contribute to the success of autoregressive pretraining for Mamba. Based on this, We introduce a pretraining strategy specifically designed for the Mamba-Transformer hybrid framework for the first time. This strategy is effective not only for the hybrid backbones but also for pure Mamba and pure Transformer backbones. We have validated the effectiveness of our approach on both 2D and 3D datasets.

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
