# OpenReview forum: "MAP: Unleashing Hybrid Mamba-Transformer Vision Backbone’s Potential with Masked Autoregressive Pretraining"
_ICLR.cc/2025/Conference — ICLR 2025 Conference Withdrawn Submission_

### Official Review · Reviewer_ij9g · 2024-10-31

**Soundness:** 3
**Presentation:** 1
**Contribution:** 2
**Rating:** 3
**Confidence:** 5

**Summary:**

The paper proposes a revised auto-regressive self-supervised training method, MAP, that is specifically designed for the Hybrid Mamba-Transformer network. Unlike the language domain, auto-regressive training underperforms masked image modeling as a transformer training method in the vision domain. However, the paper finds that it is not generalized to Mamba architecture and argues that auto-regressive training can outperform MAE with proper design and hyper-parameter tuning. Based on a lot of pilot studies, the proposed auto-regressive training, MAP, exhibits impressive performance surpassing MAE with Hybrid Mamba-Transformer on ImageNet.

**Strengths:**

- MAP training achieves reasonable improvement with Mamba-based architecture. It might be a valuable finding for researchers developing Mamba to make it better than Transformer.
- The paper presents extensive experiment results.

**Weaknesses:**

- The effectiveness of auto-regressive training has already been verified by a previous paper [A]. It significantly reduces the novelty and contribution of the paper's findings.
- Although the paper shows various findings, the findings lack general applicability and theoretical evidence. More experiments are needed to verifies the findings can be applied to general cases, which can contribute to the community.
- The proposed method, MAP, seems more like a mixture of heuristics and hyper-parameter tuning than a novel training method. It is incremental.
- Performance improvement in the 3D domain looks marginal. Considering that HybridMT3D requires additional # params and FLOPs, it is not clear that it is better than other methods.
- The writing quality is not good. It is hard to understand the main argument through the texts

[A] Autoregressive pretraining with mamba in vision, 2024

**Questions:**

- What is the major difference between [A] and MAP? All details presented in the paper are necessary to make high-performance auto-regressive training recipes?
- I think the paper's findings lack novelty and impact. Mamba+AR performance has already been demonstrated in [A]. Matching scanning orders on Mamba and AR looks like a straightforward choice.  Basic random masking with MSE loss is also the best choice for AR. Is there any novel point in MAP that was never presented in existing papers and doesn't align with MAE practices?
- Where is the definition of `HybridMH` and `HybridMT3D`? It looks like the paper's original model, but I can't find a proper description of it.
- Why is MAP super effective only in Mamba architecture, and is Hybrid needed to maximize the improvement? Is there any theoretical explanation or insight for this?
- 3D performance looks marginal. Why is MAP less effective in 3D rather than 2D? Is this improvement meaningful? Because `HybridMT3D` requires additional params and FLOPs, it looks simple computation-performance trade-off for me.

---

### Official Review · Reviewer_hAvv · 2024-10-31

**Soundness:** 3
**Presentation:** 3
**Contribution:** 3
**Rating:** 5
**Confidence:** 4

**Summary:**

This paper introduces Masked Autoregressive Pretraining (MAP) technique for hybrid Mamba-Transformer vision backbones. This method aims to address limitations of existing pretraining technique for Mamba and Transformer architectures by combining features from both Masked Autoencoding (MAE) and Autoregressive (AR) methods. The approach involves masking image patches randomly and reconstructing them in a row-by-row autoregressive manner. The author provides empirical explorations on MAP masking patterns, ratios, as well as different hybrid architecture designs. Experimental results show the improvement of using MAP on hybrid mamba-transformer architecture.

**Strengths:**

1. The motivation for designing this method is solid, and empirical results prove the effectiveness of applying MAP on hybrid mamba-transformer vision architecture across different size, comparing to MAE or AR pretraining.
2. Extensive baseline comparisons with pure transformer based model, pure mamba based model, and hybrid model from previous works.
3. Good flow of the paper with walkthrough of each component of the proposed method.

**Weaknesses:**

1. Lack of in-depth analysis on why MAP works. Although the author has included ablations on various aspect of the training and architecture design, most of them are pure tuning without analyzing the reason behind it. It would be more interesting adding some analysis on why, for example, row-wise causal decoder is working better.
3. Lack of more downstream transfer experiments on dense predictions, such as semantic segmentations on ADE20K. Existing reported experiments for 2D task is only reported on ImageNet1K classification.
3. Limited exploration on the AR scan pattern. ARM[1] explores cluster-based raster scan, which performs better than row-wise of column-wise.
4. Lack of hybrid mamba-transformer baseline with MAP applied. The comparison in 2D task lacks the baseline method with MAP. I'm not sure if the reported performance is because of the architecture change or the proposed pretraining.


[1] Autoregressive Pretraining with Mamba in Vision

**Questions:**

Refer to the weakness section.

---

### Official Review · Reviewer_bQeS · 2024-11-04

**Soundness:** 2
**Presentation:** 1
**Contribution:** 2
**Rating:** 3
**Confidence:** 5

**Summary:**

This paper introduces Masked Autoregressive Pre-training (MAP) for a Mamba, which is a method to enhance the scalability of Mamba (technically Vision Mamba, Vim) by applying Masked Autoencoder (MAE) in an autoregressive manner for the pre-training method. By applying MAP to a hybrid Mamba-Transformer architecture, the authors achieve state-of-the-art performance on 2D and 3D datasets like ImageNet-1K and MOdelNet40, surpassing existing pre-training strategies. Ablation studies support the efficacy of MAP's design choices.

**Strengths:**

- Exploring MAE-like pre-training methods on state space models like Mamba is still relatively unexplored, making this a valuable research area.
- The study reveals intriguing findings, such as the effectiveness of autoregressive pre-training for Mamba and the promising approach of integrating mask tokens into autoregressive pre-training.

**Weaknesses:**

- The paper’s contributions need to be stated more clearly, and some aspects lack novelty. While it proposes both a pre-training method -- Masked Autoregressive Pre-training (MAP) -- and a hybrid architecture (i.e., Mamba-Transformer), their individual contributions are not clearly distinguished, and introducing hybrid architecture itself is not novel. The authors begin by highlighting MAP's effectiveness by showcasing the enhancements to the Mamba-Transformer hybrid architecture (while the pilot study only employed handled Mamba -- ViM --). The reviewer suggests a more logical flow: MAP should first be tested on a standalone Mamba, followed by evaluating it on a hybrid architecture. Furthermore, most evaluations seem to focus on hybrid architectures, which obscures MAP's real aspect and results in a blending of contributions between the method and architecture. For example, there is insufficient reasoning as to why MAP would outperform existing methods like MAE and AR, along with a missing logical link explaining why MAP would be effective specifically for the hybrid architecture. The reviewer recommends that the authors distinctly separate their contributions and provide systematic support for each, suggesting that focusing on MAP would lead to more productive outcomes.
- As mentioned above, there is a lack of evidence and intuition explaining why MAP is effective for standalone Mamba or Transformer models (actually, we need the intuition as to why MAP does not fall behind MAE for Transformers). Furthermore, the rationale for using random masking (why random masking?) in Mamba-pre-training is also lacking. Without preliminary studies testing random masking in the pilot study (only the last tokens are masked), it is unclear why this approach would work and how it can be effectively combined with Mamba training because detailed explanations are absent.
- Furthermore, the motivation for introducing a hybrid architecture immediately after MAP is unclear, causing the paper’s main contributions to diverge by adding an architectural focus that may detract from emphasizing the primary contribution.
- The pilot study does not fully align with the proposed pre-training method. While it successfully indicates that autoregressive pre-training is beneficial for Mamba, it does not clearly explain why random masking is necessary. Additionally, the reviewer feels that details like the optimal scanning direction and masking ratio for tokens at the end of the sequence, while potentially informative, are not prerequisites for the pilot study. Furthermore, the pilot study fails to provide any foundational rationale for a hybrid architecture.
- In experiments, the effectiveness of MAP on non-hybrid architectures is neither impressive nor fairly compared with competing methods: 1) improvements for ViTs are minimal, and 2) competing methods like AR are not directly compared to MAP (e.g., MambaR-B/L + AR vs. MambaR-B/L + MAP in Table 8).
- Missing related works:
   - TokenUnify: Scalable Autoregressive Visual Pre-training with Mixture Token Prediction
   - MambaMIM: Pre-training Mamba with State Space Token-interpolation
   - Audio Mamba: Selective State Spaces for Self-Supervised Audio Representations
   - CMViM: Contrastive Masked Vim Autoencoder for 3D Multi-modal Representation Learning for AD classification

- Many details are missing, including the experimental setup and the baseline used for the experiments:
  - It is unclear how Mamba can handle random masking (rather than using masks only at the end of the sequence) during pre-training.
  - The baselines and experimental setups for some experiments are missing, such as the experiments in Tables 2, 3, and others.
  - The differences between the proposed method and a simple combination of masking strategy and autoregressive modeling are not clearly presented. The reviewer only notes a causal decoder.

- Minors
  - Ablation studies, such as the full results of the masking ratio of MAP, are not sufficiently presented.
  - The authors are encouraged to add more spacing after tables and figures to improve readability.
  - Found duplicated references at lines 554-559, 560-566, 657-664, 696-701.
  - In Fig. 2(b), the masking position seems somewhat strange.

**Questions:**

See above weaknesses

---

### Official Review · Reviewer_3np1 · 2024-11-07

**Soundness:** 3
**Presentation:** 3
**Contribution:** 3
**Rating:** 6
**Confidence:** 3

**Summary:**

This paper proposes a new MAE like pre-training method designed specifically for vision mamba models. The method was inspired from the difference in AR and MAE pre-training methods applied to transformers and vims, and proposed design choices, such as the scan order, masking strategies, masking ratio, etc. as well as a hybrid transformer mamba architecture. MAP shows a good gain compared with existing MAP models without a pre-training stage.

**Strengths:**

1. The paper explored comprehensively various design choices in the MAE design and did find that the changes were necessary for mamba training performance.
2. Results were showcased extensively on image classification tasks and 3D point cloud tasks.

**Weaknesses:**

1. Quite limited performance gain compared with large ViT models, even with the hybrid MMMT design, e.g. 85.9 original MAE baseline vs. the strongest 86.2 result with also a much lower throughput.
2. The original promise of mamba was on the long sequence or long context modeling, but the design in this paper does not fully demonstrate this advantage. See the question section for more discussion here.
3. If I am understanding correctly, the goal of enabling mamba pre-training was to extend the method to longer sequences. (If this is not the case, then within the ViT-supported sequence length, we could perfectly use ViT-L with various MAE variants but not MAP). In this case, some of the design choices, to some extent, were not compatible with the original mamba design, e.g. the transformer layers in the hybrid design and the transformer decoder layers. These may not work on long sequences and may prevent the context scaling as follow up.

**Questions:**

1. Related to weakness #2, does the method work also on videos. Probably yes, given the existence of video MAE methods, then in this case, are we able to support more frames than what was possible with ViT or hierarchical or hybrid transformer models?
2. Related to weakness #2, similar to videos, would object detection or segmentation, as classic testbed of MAE methods, benefit from the mamba design and use larger image resolution? The same might even apply to MLLMs, but these may be out of scope for this paper.

---

### Official Review · Reviewer_8rdL · 2024-11-07

**Soundness:** 3
**Presentation:** 3
**Contribution:** 3
**Rating:** 8
**Confidence:** 5

**Summary:**

The paper introduces **Masked Autoregressive Pre-training (MAP)**, a pre-training method designed to improve the performance of a hybrid **Mamba-Transformer** vision backbone. While **Mamba** offers advantages in long-context modeling for vision tasks, it struggles with scalability in large parameter settings. Traditional pre-training techniques like **Masked Auto Encoder (MAE)** work well for Transformers but show limited success with Mamba. The proposed MAP technique combines MAE and autoregressive pre-training to improve both Mamba and Transformer components' performance in a unified framework. Experimental results show that MAP-pretrained hybrid models outperform other techniques in both 2D and 3D tasks across various datasets, indicating that MAP is highly effective in enhancing the Mamba-Transformer hybrid model’s performance.

**Strengths:**

1. **Innovative Hybrid Pre-training Strategy**: The proposed MAP approach effectively combines MAE and autoregressive pre-training, which promisingly addresses the limitations of existing pre-training methods for Mamba-based backbones.
2. **Comprehensive Experimental Validation**: The authors validate the effectiveness of different pre-training paradigm through extensive experiments on 2D and 3D tasks (e.g., ImageNet-1K, ModelNet40, and ShapeNetPart), showing competitive results.
3. **Scalability and Adaptability**: MAP works for hybrid architectures (Mamba and Transformer) and is promising to be applied across multiple domains.
4. **Detailed Analysis and Ablation Studies**: The paper includes in-depth studies on various factors like scanning orders, masking ratios, and layer arrangements, providing insights into the specific impacts of these design choices on model performance.

**Weaknesses:**

1. **Limited Other Tasks Validation**: Most 2D vision experiments are conducted on standard IN1K datasets, I wonder if the downstream dense prediction tasks, i.e., classification, detection, could benefit from the proposed pre-training techniques.
2. **Out-of-data Citation**: The paper references preprints and arXiv versions of significant works, such as Mamba (COLM), Vision Mamba (ICML), and VMamba (NeurIPS). The authors should update these citations to their final published versions to reflect the current state of the literature.

**Questions:**

Please refer to the weakness part.

---

### Note · Authors · 2024-11-14

**Comment:**

We would like to express our gratitude to all reviewers for their constructive suggestions. We will carefully revise the manuscript and supplement the necessary experiments.

However, regarding Reviewer bQeS, who appears to have written the review using a large language model, we believe that their comments misrepresent the contributions of our paper. Our primary contribution lies in designing the MAP pretraining framework. We never claimed that designing a hybrid framework was a contribution of this paper. While designing a hybrid framework is an important issue, it is not within the scope of this study. Moreover, the reviewer seems to have overlooked several explicit statements in the paper that address these points. The misleading nature of Reviewer bQeS's comments has significantly impacted the submission experience at ICLR.

Regarding Reviewer ij9g, we have already discussed the differences between MAP and ARM. The reviewer did not explore the core components that make ARM effective for Mamba, nor did they address this issue. Additionally, MAP is designed for pretraining hybrid frameworks, not specifically for Mamba. This should be a basic understanding of our paper, but Reviewer ij9g seems unclear on this point. More importantly, as of today, ARM has not been accepted by any conference. Our comparison with ARM was intended to provide a better comparison with contemporary work.

**Withdrawal Confirmation:**

I have read and agree with the venue's withdrawal policy on behalf of myself and my co-authors.